# Constructivo GRASP para la Optimización de una Planta de Producción en la Industria Automotriz

**Sergio Cavero**
Departamento de Informática y Estadística
Universidad Rey Juan Carlos (URJC)
Madrid, España
sergio.cavero@urjc.es

**Isaac Lozano-Osorio**
Departamento de Informática y Estadística
Universidad Rey Juan Carlos (URJC)
Madrid, España
isaac.lozano@urjc.es

**Manuel Laguna**
Leeds School of Business
University of Colorado Boulder
Boulder, CO, USA
manuel.laguna@colorado.edu

## Abstract

Este artículo aborda un problema de planificación de la producción en la fabricación de asientos para automóviles. Ante la complejidad de múltiples líneas de producción y la diversidad de los productos a producir, se propone un algoritmo heurístico basado en la metaheurística GRASP. El objetivo del algoritmo es encontrar soluciones factibles al problema y minimizar los costosos cambios de configuración, manteniendo los niveles de inventario dentro de los rangos deseados. La efectividad del algoritmo se valida comparándolo con un modelo exacto, demostrando su capacidad para generar soluciones factibles y eficaces para la mayoría de los casos reales estudiados. Los resultados confirman la utilidad del enfoque GRASP como solución eficiente y adaptable, e inspiran futuras investigaciones en heurísticas más avanzadas para este problema.

## 1. Introducción

En la industria automotriz, la fabricación de asientos para automóviles presenta diferentes desafíos: la variedad de modelos, la necesidad de mantener niveles de inventario dentro de rangos específicos y la simulación de plantas de producción que involucran complejos procesos. Este artículo aborda un problema derivado directamente de la industria de fabricación de asientos para automóviles, que involucra múltiples líneas de producción y una variedad de modelos de automóviles. El desafío principal radica en determinar la secuencia óptima de moldes, cada molde puede montarse en una línea para mantener los niveles de inventario en los rangos deseados, satisfacer la demanda y minimizar al mismo tiempo el número de cambios de configuración de la planta.

En el caso de estudio analizado, la planificación se realiza actualmente mediante métodos manuales basados en la experiencia de los trabajadores y hojas de cálculo. Esto resulta en un enfoque insuficiente ante la escalabilidad productiva y los requisitos dinámicos del sector. En una primera aproximación al problema, se implementó un modelo de programación lineal entera mixta (más conocido como MIP, del inglés *Mixed-Integer Programming*) que lo formaliza matemáticamente. Sin embargo, la dependencia de herramientas de optimización comerciales con altos costes limita su aplicabilidad práctica. Por otro lado, los elevados tiempos de generación del modelo matemático y su posterior resolución eran excesivamente elevados para la empresa en algunos escenarios. Por lo tanto, esta investigación se centra en un enfoque distinto, el diseño de algoritmos heurísticos que equilibren

XVI Congreso Español de Metaheurísticas, Algoritmos Evolutivos y Bioinspirados (maeb 2025).

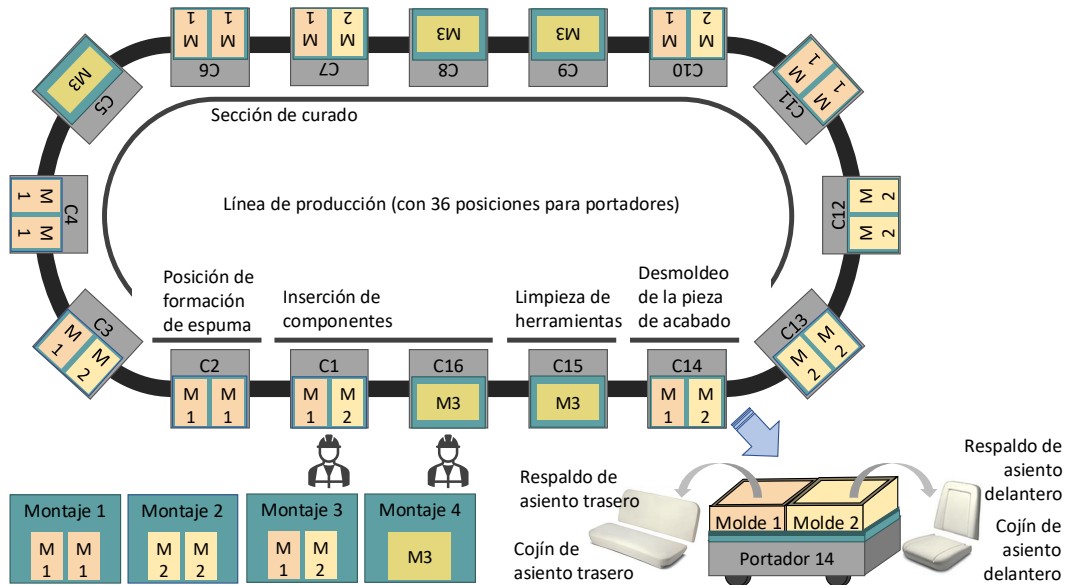

Figura 1: Representación gráfica de una planta de producción y sus componentes principales.

calidad de solución y eficiencia computacional, que, además, no requieran de la licencia de un software en concreto para poder ser ejecutados.

Dado que no es estrictamente necesario alcanzar la solución óptima global, sino obtener soluciones de calidad aceptable en tiempos computacionales razonables, se ha desarrollado un algoritmo constructivo heurístico basado en el Procedimiento de Búsqueda Adaptativa Aleatoria Codiciosa (comúnmente conocido como GRASP, por sus siglas en inglés derivadas de *Greedy Randomized Adaptive Search Procedure*). Este algoritmo está diseñado para proporcionar soluciones eficientes en instancias complejas derivadas de las operaciones reales de la empresa. El objetivo de este artículo es presentar el problema y analizar el enfoque constructivo propuesto, resaltando su capacidad para generar soluciones factibles y eficaces. Este trabajo sienta las bases para futuras investigaciones, donde se podrían explorar algoritmos heurísticos más avanzados.

En las siguientes secciones, se describe el problema (Sección 2), la literatura relevante (Sección 3), el algoritmo heurístico propuesto (Sección 4), y los resultados de los experimentos computacionales realizados para evaluar su rendimiento (Sección 5). Finalmente, se presentan las conclusiones y las líneas de investigación futuras (Sección 6).

## 2.    Presentación del problema

El problema de planificación de la producción abordado en este trabajo se sitúa en un sistema de fabricación de automóviles, concretamente en líneas de producción heterogéneas con forma circular (también denominadas como "pistas de producción", "líneas" o "pistas"). Estas líneas de producción disponen de posiciones para alojar portadores, que a su vez contienen montajes equipados con moldes para la fabricación de productos (partes de asientos). La Figura 1 ilustra una representación gráfica de la planta y sus componentes principales.

Cada línea de producción posee una capacidad productiva diferenciada, determinada por su velocidad y cantidad de portadores que puede albergar. Los portadores contienen montajes de moldes (denotados como "M1", "M2" y "M3" en la Figura 1), sujetos a restricciones de capacidad (el tamaño total de los moldes no debe superar el 100 % de la capacidad del portador) y compatibilidad (atributos como productos químicos y ángulos de inclinación). Los moldes se destinan a la producción de diversos tipos de partes de asientos, cada uno con un nivel de inventario objetivo y una demanda diaria.

El horizonte de planificación considera periodos de varios días para la demanda y franjas temporales dentro de cada día para secuenciar la producción. La capacidad de producción de cada línea varía según el número de posiciones de portadores y su velocidad. El inventario de cada parte se incrementa

con la producción diaria y se reduce con la demanda al final del día. No satisfacer la demanda se considera una solución no factible. Adicionalmente, existen restricciones de inventario superior e inferior (nivel de seguridad) para cada producto.

Una instancia del problema se define por: las características de las líneas de producción (portadores, moldes y productos); el inventario inicial; las demandas diarias; la planificación del horario de producción (días y franjas de producción); y el estado inicial de las líneas de producción. El objetivo principal es obtener una planificación factible que satisfaga la demanda. Entre las soluciones factibles se busca minimizar el número de cambios de portadores, que se denotan como "intercambios" o "cambios", y la desviación del inventario respecto a los niveles objetivo. La minimización de cambios de portadores es la función objetivo principal, mientras que las desviaciones del nivel objetivo de inventario se usan como medida de rendimiento de la producción de la planta.

Desde el punto de vista empresarial, se debe proveer a la empresa de una planificación de cada línea, indicando los portadores y moldes que se utilizarán a lo largo de todo el horizonte de planificación. Por lo tanto, la principal decisión del problema de optimización será determinar la combinación de moldes que se montarán en los portadores y la línea de producción en la que este estará asignado. Estas decisiones deben tomarse al comienzo de cada intervalo de producción. Una vez que un portador con un conjunto particular de moldes se monta en la línea antes del inicio de un intervalo, el portador produce las partes correspondientes durante todo el intervalo. Cabe mencionar que hay un límite en el número de cambios de portadores que pueden ocurrir entre dos franjas de tiempo consecutivas, pudiendo ser cero.

## 3.  Literatura

El problema de planificación de las líneas de producción en el entorno de la fabricación de asientos de coche se puede clasificar dentro de la familia de problemas de dimensionamiento discreto de lotes y planificación de máquinas heterogéneas paralelas (en inglés, *discrete lot-sizing and scheduling of parallel heterogeneous machines*). Este tipo de problemas ha sido ampliamente estudiado en la literatura debido a su importancia en la optimización de procesos industriales. Diversos enfoques y modelos han sido propuestos para abordar las complejidades inherentes a estos problemas. Concretamente, en [3] se realiza una revisión exhaustiva y una clasificación de los modelos que abordan problemas simultáneos de dimensionamiento y planificación de lotes. Esta revisión proporciona una visión general de los enfoques y técnicas utilizados en la literatura para resolver estos problemas complejos. En esta sección, se recogen algunos problemas que tienen una mayor relación con el trabajo abordado

De Matta y Guignard desarrollaron un modelo de programación de enteros mixtos para un problema de dimensionamiento de lotes y planificación en una empresa de fabricación de azulejos [4]. Su objetivo era minimizar la suma de los costes de producción, intercambios y mantenimiento de inventarios. Los costes de producción e intercambios dependían del producto y la línea, pero no del período de producción o la secuencia de producción. Utilizaron un enfoque de relajación lagrangiana para descomponer el problema en subproblemas de una sola línea, cuyas soluciones se combinaban para generar un programa a nivel de planta. Dado que este programa no garantizaba soluciones factibles, desarrollaron una heurística para lograr la factibilidad.

Jans y Degraeve propusieron un modelo y un procedimiento de solución para un problema de planificación de la producción en un fabricante de neumáticos [8]. Los moldes y calentadores producen neumáticos en grandes series de ciclos de producción. El problema resultante de dimensionamiento de lotes es una extensión del problema estándar de dimensionamiento y planificación de lotes discretos [13]. Las complicaciones adicionales incluyen tiempos de inicio de producción generales, múltiples máquinas alternativas con diferentes configuraciones, múltiples recursos (moldes y calentadores) y gestión de inventarios.

Güngör, Unal y Taşkın abordaron recientemente un problema similar en la industria automotriz, donde los moldes se utilizan para fundir ruedas de aluminio [7]. Su objetivo principal era aumentar la utilización de los recursos de producción minimizando el número de cambios. Ignoraron los tiempos de configuración y desmontaje y las restricciones de flujo de inventario porque podían tratar cada semana de producción de manera independiente. En este artículo se proponen dos formulaciones de programación de enteros mixtos, demostrando que el problema es $\mathcal{NP}$-difícil y ofreciendo una aproximación heurística.

Colmenar, Laguna y Martín-Santamaría abordan un problema similar, también en la industria automovilística, en el que, en este caso, se producen piezas metálicas para asientos de coche [2]. El objetivo del problema es minimizar los tiempos de cambio en líneas de producción en un entorno complejo caracterizado por máquinas heterogéneas paralelas y una alta variedad de piezas.

A pesar de la gran variedad de problemas estudiados, ninguno aborda un problema tan específico como el presentado en este trabajo, que contempla numerosas restricciones derivadas del cliente. Esto convierte al problema en un gran reto. Aunque se pueden extraer ideas y estrategias de la literatura para abordar restricciones concretas, ninguna resuelve este problema concreto. Adicionalmente, desde el punto de vista de la formulación matemática, existe una limitación con el alto coste de la licencia comercial. Es por ello, que este trabajo propone una heurística específica para resolver este problema de manera efectiva.

## 4. Propuesta algorítmica

El objetivo de esta sección es presentar un algoritmo constructivo aleatorizado inspirado en GRASP [5, 6] con la finalidad de encontrar soluciones factibles para el problema estudiado, minimizando el número de intercambios de portadores y, de manera secundaria, mejorar los indicadores de rendimiento relacionados con el inventario. Se recuerda que una solución factible es aquella en la que se satisfacen todas las demandas de los clientes. Esta sección presenta, en primer lugar, la representación de la solución (Sección 4.1) y luego el algoritmo propuesto (Sección 4.2).

### 4.1. Representación de la solución

Para presentar el algoritmo propuesto, es necesario definir la representación formal de una solución y las operaciones fundamentales para modificarla.

Inicialmente, se asume la creación de un catálogo de montajes compatibles con cada pista de producción $r \in R$, considerando restricciones de capacidad y compatibilidad. Además, se define $K_r$ como el conjunto de portadores compatibles con la pista $r$, considerando que no todos los moldes ni todos los montajes son compatibles con todas las pistas. Esta simplificación permite evitar la necesidad de modelar explícitamente todas las restricciones inherentes a la combinación de moldes, montajes y portadores a lo largo del proceso de producción, centrando el algoritmo en la gestión de portadores en cada pista.

Una solución $Q$ se representa como un conjunto de listas de portadores, donde cada lista $Q_{rts}$ contiene los portadores $k \in K_r$ asignados a la pista $r$ en la franja horaria $s \in S_t$ del día $t \in T$:

$$Q = \{Q_{rts} : r \in R, t \in T, s \in S_t\},$$

$$Q_{rts} = [k_1, k_2, \ldots, k_p] \text{ donde } k_i \in K_r \text{ para } i = 1, \ldots, p,$$

donde $p$ es el identificador de un portador.

La cardinalidad $|Q_{rts}|$ es igual al número de puestos en la pista $r$. Inicialmente, $Q$ refleja el estado de partida de las pistas.

Se definen dos operaciones básicas para modificar una solución $Q$: "Añadir" ($k_{\text{add}}^{rts}$) y "Quitar" ($k_{\text{drop}}^{rts}$). La operación "Añadir" asigna un portador $k$ a una posición disponible en la pista $r$ durante la franja $s$ del día $t$, insertándolo en la lista $Q_{rts}$. La operación "Quitar" retira un portador de la pista $r$ en la franja $s$ del día $t$, eliminándolo de $Q_{rts}$. La transición entre soluciones se representa de la siguiente manera:

$$Q_{rts} \leftarrow Q_{rts}^o \cup \{k_{\text{add}}^{rts}\} \setminus \{k_{\text{drop}}^{rts}\}.$$

Por simplicidad, se asume que el efecto de una operación en una franja horaria $s$ del día $t$ persiste en todas las franjas posteriores hasta el final del horizonte de planificación, a menos que se aplique una operación posterior que involucre el portador añadido o retirado. Además, es importante considerar el número de cambios de portadores permitidos en cada franja de tiempo.

La calidad de una solución $Q$ se evalúa mediante dos componentes principales: el número de intercambios y la desviación con respecto a los inventarios deseados. El número de intercambios

es la función objetivo principal a minimizar y se calcula contabilizando las operaciones $k_{add}^{rts}$ y $k_{drop}^{rts}$ a lo largo del horizonte de planificación. La desviación de inventarios se utiliza como métrica de rendimiento auxiliar, midiendo la diferencia entre los inventarios resultantes y los niveles objetivo. Adicionalmente, se verifica la factibilidad de la solución comprobando si se satisfacen todas las demandas de producción.

Para optimizar la eficiencia, se implementan actualizaciones incrementales al aplicar las operaciones. Al añadir o quitar un portador, solo se recalculan los inventarios de las partes afectadas por el cambio, desde el día $t$ en adelante. A pesar de estas optimizaciones, es importante reconocer que la ejecución de cada operación sigue siendo computacionalmente costosa, requiriendo la actualización de $Q$ y la revaluación de la función objetivo y la factibilidad. El diseño eficiente de estas operaciones y evaluaciones incrementales es crucial para el rendimiento de los algoritmos de optimización.

## 4.2. Algoritmo constructivo basado en GRASP

El procedimiento de Búsqueda Voraz, Aleatorizado y Adaptativo, más conocido por su acrónimo GRASP (*Greedy Randomized Adaptive Search Procedure*), es una metaheurística iterativa ampliamente utilizada en diferentes problemas de optimización [9, 11, 12]. Cada iteración de GRASP se centra en dos fases: construcción y mejora. En la fase de construcción, GRASP genera una solución factible mediante un proceso voraz y aleatorizado. Aunque GRASP típicamente incluye una fase mejora con una búsqueda local, en este trabajo el enfoque se centra exclusivamente en la fase constructiva, generando múltiples soluciones durante un tiempo máximo establecido y seleccionando la mejor de entre ellas.

Un elemento central de la fase constructiva de GRASP es la función voraz. En el algoritmo constructivo GRASP que se propone en este trabajo, se emplean dos funciones voraces distintas, una para la selección de portadores a añadir y otra para la selección de portadores a retirar. La función voraz para añadir se evalúa el impacto de incorporar un portador específico a la solución en construcción, estimando su contribución a la reducción de la escasez de partes. Concretamente, cuantifica el número de partes en escasez que un portador candidato puede producir. Por otro lado, la función voraz para la retirada evalúa el impacto de retirar un portador, cuantificando el número de partes en escasez que dejarán de producirse al retirar el portador.

Basándose en estas dos evaluaciones voraces, esta propuesta GRASP construye dos listas de candidatos restringidas (RCL): RCL$_{add}$ para añadir portadores y RCL$_{drop}$ para eliminar portadores. La RCL$_{add}$ contiene los portadores mejor evaluados para ser añadidos, es decir, aquellos que ofrecen la mayor reducción potencial en la escasez de partes según la función voraz de adición. La RCL$_{drop}$ contiene los portadores mejor evaluados para ser retirados, es decir, aquellos que, según la función voraz de retirada, producen el menor número de partes en escasez (minimizando el impacto negativo de la retirada). La aleatoriedad se introduce al seleccionar un portador para añadir desde la RCL$_{add}$ y un portador para retirar desde la RCL$_{drop}$, ambos de forma aleatoria dentro de sus respectivas listas.

El algoritmo constructivo itera hasta alcanzar una solución factible o cuando se exceda el tiempo máximo que la compañía establece para obtener una solución, realizando intercambios de portadores únicamente si contribuyen a disminuir la escasez de partes. Concretamente, el pseudocódigo del algoritmo constructivo GRASP propuesto se detalla en el Algoritmo 1.

El proceso comienza inicializando una solución (Paso 2) y, en cada iteración, identifica las partes con escasez (Paso 4). Luego, genera una lista de portadores candidatos (K$_{candidatos}$) que producen partes en escasez y que son candidatos para ser añadidos (Paso 5). Para cada portador candidato en K$_{candidatos}$, la función voraz de añadir calcula el número de partes en escasez (Paso 9). Se construyen RCL$_{add}$ (Paso 11) y RCL$_{drop}$ (Paso 17) basándose en los valores voraces de los portadores candidatos a añadir y a retirar, respectivamente. Se selecciona aleatoriamente un portador de RCL$_{add}$ para ser añadido (Paso 12). A continuación, se determina la pista y la franja horaria de un día ($r^*$, $s^*$ y $t^*$ respectivamente) en la que se añadirá el portador (Paso 13). Concretamente, se selecciona aquella combinación que minimiza la escasez, evitando añadirlo en un momento temporal cuyo aporte a la solución vaya a ser insignificante (por ejemplo, si se produce después de que no se haya podido satisfacer la demanda) o bien cuando no es posible realizar cambios de portadores. Simultáneamente, se selecciona aleatoriamente un portador de RCL$_{drop}$ para ser retirado de la pista donde se realizará la adición (Paso 18). El intercambio se realiza en la primera franja horaria válida que permita reducir la escasez (Paso 19), persistiendo el cambio hasta el final del horizonte de planificación.

---

**Algorithm 1** Algoritmo constructivo GRASP

---

1: Entradas: $K$, $R$, $T$, $S_t$, demandas, $Q^0$, $\alpha \in [0,1]$.
2: $Q \leftarrow Q^0$
3: **while** $Q$ no es factible **and** no se alcanza el tiempo límite **do**
4:     Calcular escasez $z$ para $Q$.
5:     $K_{\text{candidatos}} \leftarrow$ portadores que producen partes en escasez $z$.
6:     **if** $K_{\text{candidatos}} = \emptyset$ **then**
7:         **return** $Q$.
8:     **end if**
9:     Evaluar $K_{\text{candidatos}}$ con FunciónVoraz$_{\text{add}}$.
10:     Obtener $v_{\text{mejor}}^{\text{add}}$ y $v_{\text{peor}}^{\text{add}}$.
11:     RCL$_{\text{add}} \leftarrow \{k \in K_{\text{candidatos}} \mid \text{FunciónVoraz}_{\text{add}}(k) \geq v_{\text{mejor}}^{\text{add}} - \alpha \cdot (v_{\text{mejor}}^{\text{add}} - v_{\text{peor}}^{\text{add}})\}$.
12:     Seleccionar $k_{\text{add}} \in$ RCL$_{\text{add}}$ aleatoriamente.
13:     Determinar $r^*$, $s^*$ y $t^*$ para $k_{\text{add}}$ que reduzca escasez $z$.
14:     $K'_{\text{candidatos}} \leftarrow$ portadores en $Q_{r^*t^*s^*}$ que se pueden retirar.
15:     Evaluar $K'_{\text{candidatos}}$ con FunciónVoraz$_{\text{drop}}$.
16:     Obtener $v_{\text{mejor}}^{\text{drop}}$ y $v_{\text{peor}}^{\text{drop}}$.
17:     $RCL_{\text{drop}} \leftarrow \{k \in K'_{\text{candidatos}} \mid \text{FunciónVoraz}_{\text{drop}}(k) \leq v_{\text{peor}}^{\text{drop}} + \alpha \cdot (v_{\text{mejor}}^{\text{drop}} - v_{\text{peor}}^{\text{drop}})\}$.
18:     Seleccionar $k_{\text{drop}} \in RCL_{\text{drop}}$ aleatoriamente.
19:     $Q_{r^*t^*s^*} \leftarrow Q_{r^*t^*s^*} \cup \{k_{\text{add}}^{r^*t^*s^*}\} \setminus \{k_{\text{drop}}^{r^*t^*s^*}\}$.
20:     Actualizar $Q$.
21: **end while**
22: **return** $Q$

---

El algoritmo finaliza cuando se encuentra una solución factible (Paso 6), *i.e.*, sin escasez, cuando $K_{\text{candidatos}}$ está vacía (Paso 22), indicando que no hay portadores disponibles que puedan reducir la escasez o cuando se excede el límite de tiempo. Es importante reconocer que, inherentemente, el constructivo propuesto no garantiza la factibilidad en todos los casos (Paso 7), siendo necesaria la posible implementación de procedimientos de reparación en trabajos futuros como se ha planteado en trabajos similares por medio de procedimientos de búsqueda local [1].

Para mantener la claridad del pseudocódigo, se han simplificado ciertos detalles de la implementación. Es importante señalar que durante la ejecución pueden presentarse dos situaciones críticas no explícitas en el algoritmo: (1) que para un portador seleccionado $k_{\text{add}}$ no exista ninguna combinación válida de franja, pista y día ($r^*$, $s^*$, $t^*$) que cumpla con las restricciones del problema (Paso 13), o (2) que no se encuentre ningún portador candidato para ser eliminado de la pista (Paso 14). En ambos casos, el algoritmo retrocedería para seleccionar otro portador de la RCL$_{\text{add}}$ (iría al Paso 11). Si se agotan todos los candidatos de la RCL$_{\text{add}}$ sin encontrar una combinación viable, la solución se declarará no factible.

Por último, también es importante indicar que hay un único parámetro $\alpha$ que se utiliza para generar ambas listas RCL$_{\text{add}}$ y RCL$_{\text{drop}}$ (Pasos 11-17). En un trabajo futuro, podría considerarse el manejo de dos parámetros $\alpha$ distintos para controlar de manera independiente el tamaño y la calidad de las listas de candidatos para añadir y retirar portadores, respectivamente, lo que podría conducir a una mejor exploración del espacio de soluciones.

## 5. Experimentos

Toda la experimentación se ha desarrollado en un servidor con las siguientes características: procesador AMD EPYC 7282 (2.8 GHz), 32 cores, 8 GB de memoria RAM y Ubuntu Server 20.04. El algoritmo heurístico ha sido implementado en Java 21 utilizando el *framework* de desarrollo MORK [10], mientras que el modelo matemático se ha codificado en Python 3.9 y resuelto con Gurobi 11.0.3. Para evaluar el rendimiento de ambos enfoques, se han utilizado 60 instancias reales proporcionadas por la industria, debidamente anonimizadas para preservar la confidencialidad de los datos.

Los experimentos realizados persiguen diversos objetivos. El primero busca determinar la configuración paramétrica que maximice el número de soluciones factibles obtenidas. Esta priorización se

justifica porque, en el contexto real de la industria para la cual se elabora esta investigación, encontrar una solución factible resulta más crítico que minimizar cualquier objetivo. La Figura 2 muestra, para cada valor de $\alpha$ en el rango $[0.00, 0.50]$, el número de soluciones sin escasez que se obtienen. Los valores de $\alpha$ superiores a 0.50 han sido omitidos del análisis, dado que, experimentalmente, se comprobó que ninguna configuración con estos valores logró generar soluciones sin escasez.

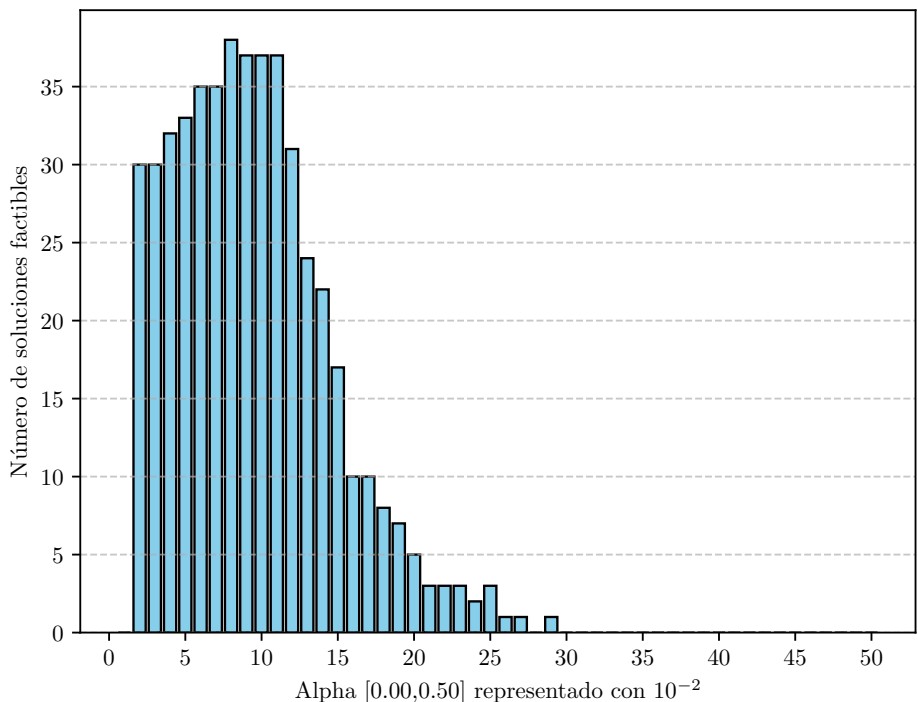

Figura 2: Número de soluciones factibles según valor $\alpha$.

Como puede observarse en la figura, el intervalo $[0.01 - 0.15]$ concentra los valores de $\alpha$ que generan mayor cantidad de soluciones factibles. Específicamente, $\alpha = 0.07$ alcanza el máximo rendimiento, produciendo soluciones factibles para 38 de las 60 instancias evaluadas. Por lo tanto, este valor de $\alpha$ ha sido seleccionado para la comparación posterior con el algoritmo exacto.

El segundo experimento analiza la convergencia del algoritmo a lo largo del tiempo de ejecución. Cabe destacar que la empresa estableció como requisito operativo obtener soluciones en menos de 10 minutos. Por esa razón, se fijó este valor como tiempo máximo de ejecución para todas las instancias. La Figura 3 muestra la evolución del promedio de intercambios (función objetivo) durante el tiempo de ejecución.

Como puede observarse en la gráfica, la reducción en el número promedio de intercambios presenta tres fases diferenciadas: una disminución significativa durante los primeros 150 segundos, seguida de una reducción más moderada hasta aproximadamente los 400 segundos, tras lo cual la convergencia se estabiliza y apenas se producen mejoras adicionales en la función objetivo hasta completar los 600 segundos (10 minutos) de ejecución.

Finalmente, en un último experimento se compara el rendimiento del método exacto (Gurobi) frente a la propuesta constructiva GRASP. Cabe señalar que el análisis se centra en las 38 instancias para las cuales se encontró una solución factible. Los resultados obtenidos se presentan en la Tabla 1. Las métricas reportadas son: Intercambios (promedio entre todas las instancias), Inventario (desviación respecto a los inventarios objetivo), Tiempo (s) y #Óptimo (número de soluciones óptimas encontradas).

La Tabla 1 evidencia que Gurobi obtiene la solución óptima en 16,16 segundos. En contraste, el método constructivo GRASP, con un tiempo de cómputo limitado a 600 segundos, se desvía un 7,30 % del óptimo, pero identifica 25 soluciones óptimas de las 38 instancias consideradas. En

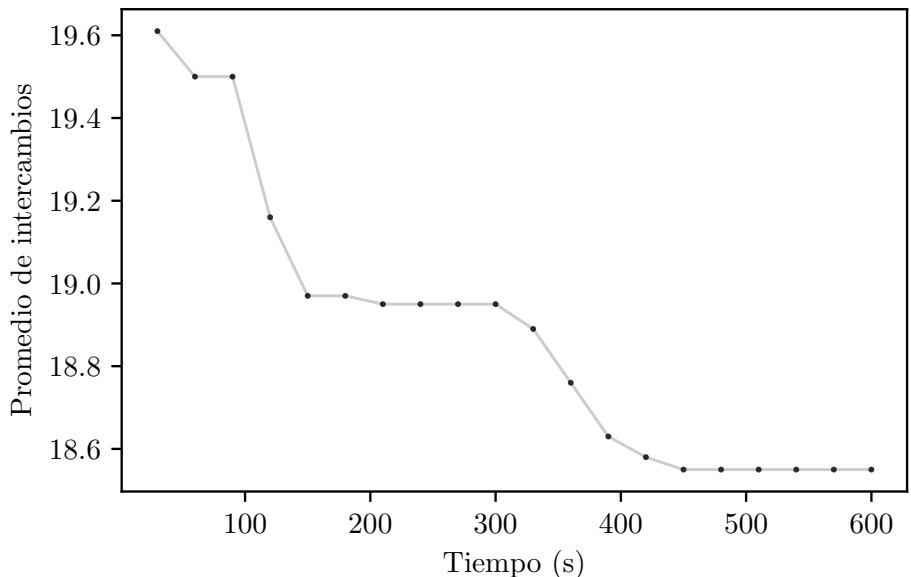

Figura 3: Evolución temporal del promedio de intercambios necesarios para todas las instancias.

Tabla 1: Resultados de Gurobi frente a la propuesta sobre las instancias que encuentran solución.

| Algoritmo | Intercambios | Inventario | Tiempo (s) | #Óptimo |
|---|---|---|---|---|
| Gurobi | 16,87 | 987,12 | 16,16 | 38 |
| Constructivo GRASP | 18,55 | 1069,63 | 600,00 | 25 |

promedio, Gurobi requiere 16,87 intercambios frente a 18,55 de GRASP. Es importante destacar que, en la práctica industrial, dos intercambios adicionales podrían implicar un coste inferior a la implementación de la solución exacta.

## 6. Conclusión

En este artículo, se ha abordado un problema de planificación de la producción en la industria de fabricación de asientos para automóviles, caracterizado por líneas de producción heterogéneas y una gran variedad de productos. Ante la complejidad del problema, derivado directamente de la industria, así como las limitaciones de los modelos exactos en términos de coste computacional y requisitos de software, se ha propuesto un algoritmo heurístico constructivo basado en la metaheurística GRASP.

Los resultados experimentales obtenidos con 60 instancias reales proporcionadas por la industria demuestran la eficacia del algoritmo GRASP propuesto, obteniendo soluciones factibles para 38 de las 60 instancias en un tiempo máximo de ejecución de 600 segundos, cumpliendo así con los requisitos operativos de la empresa. De la comparación con el modelo exacto resuelto por Gurobi se puede concluir que el constructivo GRASP resulta una alternativa eficiente en tiempo computacional y sin la dependencia de software comercial.

Si bien el algoritmo constructivo GRASP no garantiza la optimalidad global ni la factibilidad en todos los casos, representa un punto de partida sólido para futuras investigaciones en este campo. Como líneas de investigación futuras, se plantea el desarrollo e integración de fases de búsqueda local, la implementación de mecanismos de reparación, o el planteamiento de otros criterios voraces para el constructivo propuesto.

## Agradecimientos y declaración de financiación

Esta investigación ha sido parcialmente financiada mediante subvenciones: PID2021-125709OA-C22, financiado por MCIN/AEI/10.13039/501100011033; "Proyectos Impulso de la Universidad Rey Juan Carlos 2024" con referencia 2024/SOLCON-135988; CIRMA-CM Ref. TEC-2024/COM-404 financiado por la Comunidad Autónoma de Madrid; TSI-100930-2023-3 (MCA07) financiado por Ministerio para la Transformación Digital y de la Función Pública. También, los autores quieren agradecer por haberles permitido realizar esta investigación y su difusión a Better Business Analytics, consultora con la que colaboran los autores.

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
