# OpenReview forum: "Constructivo GRASP para la Optimización de una Planta de Producción en la Industria Automotriz"
_MAEB/2025/Congreso — MAEB 2025_

### Official Review · Reviewer_8TS3 · 2025-03-18
**Aplicación de un método bien conocido a un problema específico, sin mucho interés científico**

**Rating:** 2
**Confidence:** 5

**Review:**

Los autores presentan una implementación del algoritmo GRASP sin considerar optimización local, solo la fase de construcción para su aplicación a una línea de producción de asientos de automoción en el marco de un proyecto con industria.
El artículo está bien escrito con una estructura adecuada. Se presenta el método utilizado de forma clara, así como una serie de experimentos que han hecho para validarlo.
De todos modos, el artículo no deja de ser la aplicación de un método estándar a un problema específico. De hecho, científicamente no aporta ninguna novedad. Por otra parte, los resultados que se presentan son dos gráficos para asegurar el cumplimiento de los requisitos del cliente y una pequeña tabla comparando el método con uno exacto. El resultado de la comparación no es muy favorable para los autores, que lo justifican aludiendo a cuestiones de la industria en sí.
En definitiva, un artículo bien escrito pero que aporta muy poco. Solamente un ejemplo más de aplicación de un algoritmo a un entorno específico.

---

### Official Review · Reviewer_GcmY · 2025-03-19
**Metaheurístico para la optimización de líneas de producción**

**Rating:** 3
**Confidence:** 5

**Review:**

El paper propone un algoritmo heurístico constructuvo que se inspira en el metaheurístico GRASP para resolver el problema de planificación de la producción. Compara el approach propuesto con un modelo de programación lineal entera mixta en 38 instancias.

La motivación de proponer un heurístico para resolver este problema viene dado por la dependencia de herramientas comerciales de alto coste  como Gurobi que se utilizan para resolver modelos como el MIP y el elevado coste computacional de resolver el problema con un método exacto. Aunque el primero puede ser cierto, en las instancias propuestas Gurobi necesita 16 segundos para resolverlos (bastante menos que que el tiempo máximo considerado para el heurístico). Hecho de menos más información sobre la complejidad de las instancias como el número de portadores, inventarios etc...

---

### Official Review · Reviewer_zXqQ · 2025-03-19
**Este trabajo propone un método heurístico GRASP para la resolución de un problema de planificación de linea de producción en la industria automotriz.**

**Rating:** 2
**Confidence:** 4

**Review:**

La motivación y presentación del contexto del problema son adecuadas. Se explican las particularidades del problemas caracterizado por un amplio conjunto de restricciones que lo hacen más complejo. También se aporta una contextualización del  escenario industrial donde existe este tipo de problemas. La explicación de las componentes del algoritmo GRASP diseñadas para resolver el problema también es clara.

Las dos limitaciones más importantes de esta contribución están en el diseño del algoritmo cuyos operadores no garantizan la obtención de soluciones factibles. Tendría más sentido considerar operadores que alteren la solución actual sin perder la condición de factibilidad o introducir métodos de reparación, una posibilidad que se posterga en el trabajo para investigaciones futuras.

El otro elemento débil de la contribución es el marco experimental definido que es técnicamente poco riguroso e incompleto. La decisión de comparar los resultados del método MIP y el algoritmo introducido únicamente para las 38 instancias para las cuales el segundo método es capaz de encontrar soluciones factibles es cuestionable y no aparece fundamentada en el paper. Tampoco se menciona en el paper cuántas ejecuciones de GRASP se realizaron para cada una de las 60 instancias. Da la impresión que se realizó una única ejecución para cada instancia (y valor de alpha). En ese sentido la información disponible en las figuras 2 y 3 y la Tabla 1 es claramente insuficiente para determinar el valor relativo del algoritmo introducido. También faltaría incluir la comparación con algun otro método heurístico o contar con el valor de una solución propuesta por los humanos para cada una de las 60 instancias, lo cual permitiría evaluar la ventaja aportada por el método introducido.

En resumen, se trata de una aproximación interesante a un problema muy difícil pero los resultados contenidos en el trabajo indican que la investigación se encuentra aún en una fase inicial y se necesitan mejoras al algoritmos y la concepción de un marco experimental robusto para evaluar la validez del método.

---

### Decision · Program_Chairs · 2025-03-20

**Decision:**

Accept

**Comment:**

De cara a preparar la versión camera ready, sugerimos a los autores que revisen detenidamente los comentarios de los revisores y que traten de mejorar el manuscrito según las recomendaciones de los revisores.